# Beyond the single-site approximation modeling of electron-phonon coupling effects on resonant inelastic X-ray scattering spectra

Krzysztof Bieniasz,[1, 2, *] Steven Johnston,[3] and Mona Berciu[1, 2]

[1] *Quantum Matter Institute, University of British Columbia, Vancouver, British Columbia, Canada V6T 1Z4*
[2] *Department of Physics and Astronomy, University of British Columbia, Vancouver, British Columbia, Canada V6T 1Z1*
[3] *Department of Physics and Astronomy, University of Tennessee, Knoxville, Tennessee 37996, USA*
(Dated: August 27, 2021)

Resonant inelastic X-ray scattering (RIXS) is used increasingly for characterizing low-energy collective excitations in materials. RIXS is a powerful probe, which often requires sophisticated theoretical descriptions to interpret the data. In particular, the need for accurate theories describing the influence of electron-phonon (*e*-p) coupling on RIXS spectra is becoming timely, as instrument resolution improves and this energy regime is rapidly becoming accessible. To date, only rather exploratory theoretical work has been carried out for such problems. We begin to bridge this gap by proposing a versatile variational approximation for calculating RIXS spectra in weakly doped materials, for a variety of models with diverse *e*-p couplings. Here, we illustrate some of its potential by studying the role of electron mobility, which is completely neglected in the widely used local approximation based on Lang-Firsov theory. Assuming that the electron-phonon coupling is of the simplest, Holstein type, we discuss the regimes where the local approximation fails, and demonstrate that its improper use may grossly *underestimate* the *e*-p coupling strength.

## I. INTRODUCTION

Resonant inelastic X-ray scattering [1] is being increasingly used to study electron-phonon (*e*-p) interactions in quantum materials [2–13]. This application is facilitated by improvements in both experimental resolution [1, 14] and in our understanding of how the scattering process generates lattice excitations [3–5, 8, 15–17].

The most commonly used framework for analyzing phonon excitations in RIXS data is the atomic limit (or single-site) approximation developed by Ament *et al.* [15], which first established the theoretical connection between phonon excitation intensities and the *e*-p coupling constant. Originally developed with Cu *L*-edge measurements of the high-T$_c$ cuprates in mind, this approximation completely neglects charge fluctuations during all stages of the RIXS process and effectively treats the system as a set of isolated sites whose charge density couples to local phonon modes (see Sec. II D). This treatment was motivated by the expectation that electron correlations localize carriers in the initial and final states, while the strong core-hole potential in the intermediate state conspires with its short lifetime to confine the excited valence electron to the site where it is created. With these approximations, the RIXS cross-section can be evaluated exactly [15], resulting in an analytic expression suitable for fitting experimental data [6, 10–12]. A key result is that the ratio of intensities of successive phonon excitations are related in a one-to-one manner to the ratio $M/\Gamma$ between the *e*-p coupling $M$ and the inverse core-hole lifetime $\Gamma$.

As we show here, the accuracy of this single-site model depends significantly on the degree to which the excited valence electron is localized in the intermediate state. This

observation poses a problem for the community. Many studies of phonon excitations are conducted at the O *K*-edge, which involves an oxygen $1s \to 2p$ transition. The oxygen $1s$ core level is relatively shallow, resulting in a weaker core hole potential and a longer core hole lifetime (small $\Gamma$) [4, 8, 10]; both of these aspects favor intermediate states where the valence electron explores the neighborhood of the core-hole site instead of being localized at that one site. Furthermore, recent studies have addressed lightly doped band insulators [10], and doped cuprates [11, 12], where one or more of the states involved in the scattering process are delocalized, and yet these studies have utilized the localized model to analyze the data. To better understand the implications of these studies, we must determine how the itineracy of the electrons affects the RIXS intensity.

There have been several attempts to extend the single-site model to more general cases. For example, early efforts focused on extending the approach to small clusters using exact diagonalization (ED) [4, 8]. While these models are only able to retain a limited subset of phonon modes, they do capture the effects of electron itineracy over a few unit cells, and produce results in qualitative agreement with the single-site model (no quantitative comparison of the intensities predicted by each model has been made). More recently, Devereaux *et al.* [16] developed a model for the RIXS cross-section for a system of itinerant electrons using perturbation theory. This approach, while general, currently only includes the lowest order diagrams responsible for the single-phonon excitations. As such, its predictions cannot be directly compared against analyses based on the single-site model, which generally revolve around multi-phonon processes. Finally, Geondzhian and Gilmore [18] have shown that even within the single-site approximation, either adding coupling to a second phonon mode and/or allowing for a different phonon frequency in the intermediate state considerably affect the quantitative

---
* krzysztof.t.bieniasz@gmail.com

interpretation of the RIXS spectra.

It is important to mention also that Geondzhian and Gilmore [17] have questioned whether the intensity of the phonon excitations should be attributed solely to the coupling between the valence electrons and the lattice. Instead, they propose that the interaction should be viewed as an exciton-lattice coupling, with an additional interaction between the core hole and the lattice. We briefly discuss below the importance of such excitonic effects.

In this paper we propose a new, versatile, accurate, and numerically efficient approach for studying the effects of $e$-p coupling on RIXS spectra in band insulators. Our new approach removes most limitations inherent to the atomic limit approximation of Ref. 15 because it is based on a variational method called the Momentum Average (MA) approximation [19, 20]. Its variational nature comes from the fact that this approximation constrains the possible carrier-plus-phonons configurations to a subset that we believe is most relevant for the problem at hand. We can then verify the accuracy of the specific choice for what is retained by increasing the variational space of allowed configurations and checking if convergence has been achieved or not. This MA approach has been used very successfully to study single polarons and single bipolarons in infinite systems, at all coupling strengths, and in a variety of models with $e$-p couplings. Beside its success in dealing with the simplest Holstein coupling, the accuracy of MA has been validated for models with non-trivial diagonal $g(\boldsymbol{q})$ couplings, e.g., breathing-mode phonon couplings [21] and for off-diagonal models with $g(\boldsymbol{k}, \boldsymbol{q})$ couplings [22–24] (the latter cannot even be meaningfully studied in the single-site, atomic limit). MA has also been generalized to the study of $e$-p couplings to multiple phonon modes [25], to models involving multiple electronic bands [26, 27], and to $e$-p couplings beyond the linear approximation [28, 29]. Any combinations of the above features are straightforward to implement, e.g. $g(\boldsymbol{k}, \boldsymbol{q})$ couplings in models with multiple electronic bands [30–32] or a mix of $g(\boldsymbol{q})$ and $g(\boldsymbol{k}, \boldsymbol{q})$ couplings [33, 34]. Especially relevant for the consideration of RIXS spectra was the generalization of MA to models that include disorder [35] (the core-hole attraction is a very simple form of disorder). In this latter context, the accuracy of MA for predicting polaron spectra in the presence of attractive potentials like that due to the core-hole was validated by comparison with state-of-the-art, unbiased numerical methods in Ref. [36]. In all the work described above, the phonons were assumed to be dispersionless, Einstein modes. The generalization of MA to deal with dispersive optical phonons has been achieved very recently [37], and can thus be added to the list of cases that can be treated with MA. Insofar as single polarons and bipolarons are concerned, MA has been shown to have good quantitative accuracy everywhere in the parameter space except in the strongly adiabatic limit. Ref. [38] recently overcame this last limitation, however, by implementing a numerical procedure that allows the inclusion of an arbitrary number of configurations within the variational space.

Being based on MA, our approach for modeling RIXS spectra inherits all the capabilities listed above, allowing the investigation of a large variety of models with $e$-p coupling within a unified framework. Its only current limitation is to the study of insulators or very weakly doped materials, due to the fact that MA has not yet been extended to systems with finite carrier concentrations.

As an aside, it is also useful to emphasize that even though all the discussions here are focused on electron-phonon couplings, the same formalism can be applied to study RIXS spectra for systems with electron-magnon interactions. Indeed, our variational MA approach produces single spin-polaron dispersions in excellent agreement with Exact Diagonalization calculations for several such models (see, for instance, Refs. [39] and [40]).

The effort of implementing this new approach is only worthwhile, however, if the difference between its predictions and those of the more basic single-site approximation are considerable. Here, we demonstrate that this is the case (and use it as an opportunity to explain the general framework, which can then be generalized to all the other cases mentioned above) by investigating the fundamental question of when and how is the itinerancy of the electron in the intermediary states relevant to RIXS spectra. Specifically, we consider the case where a core electron is excited into an otherwise empty valence band, where it is free to interact with phonons. This specific problem is relevant, for example, to recent O $K$-edge experiments on $SrIrO_3/SrTiO_3$ heterostructures, where the core $1s$ electrons of the $SrTiO_3$ layers are excited into a nearly empty band [10]. (In that case, the core electron interact with the LO4 optical phonon branch, which can be approximately modeled using a $\Omega \approx 100$ meV Einstein phonon.) By comparing our results with those obtained from the atomic limit approximation we can highlight the role played by the width of the valence band, as well as the dimensionality and symmetry of the underlying lattice.

Our MA method recovers the results of the single-site model in the atomic limit (when the bandwidth of the valence band is set to zero). It also has access to multi-phonon excitations, which allows us to make quantitative comparisons between localized and itinerant cases. Using this framework, we show that while the single-site approximation captures the phonon excitations of an itinerant system qualitatively, it may significantly *underestimate* the $e$-p coupling strength. We also demonstrate that electron mobility in the intermediate state produces a momentum dependence in the intensity of the phonon excitations, even when both the underlying $e$-p coupling, and the phonon dispersion, are momentum independent. These results should be kept in mind when using the single-site model to do quantitative RIXS analysis.

The way to understand the effect of all the various features of a model with general $e$-p coupling is to add them one by one and see when and why they are relevant, i.e. when their addition leads to a quantitatively significant change to the predicted RIXS spectra. Once this knowl-

edge is collected, it will be possible to know how much detail needs to be included in a model, depending on the specific parameters of the system studied with RIXS. For example, the example chosen here will show that if the core-hole potential is very large, then the atomic limit approximation is adequate, but it becomes less so as the core-hole potential becomes smaller, in which case the MA approach must be used if accurate estimates are desired. We emphasize again that all the generalizations mentioned above can be implemented within the same framework; however, to keep the length of this paper reasonable, we will present other generalizations elsewhere.

The paper is organized as follows: In Sec. II we introduce the methodology of our research. In particular, Sec. II A presents the lattice Holstein model and the basics of the RIXS theoretical framework, Sec. II B discusses the role of coupling of the core hole to the lattice, Sec. II C outlines the implementation of the variational Momentum Average method to the calculation of the RIXS cross-section, and Sec. II D presents the Lang-Firsov solution in the atomic limit, for completeness. From there, we proceed to the numerical results in Sec. III, where we present several interesting theoretical consequences of electron mobility for the RIXS cross-section. Finally, in Sec. IV, we conclude the paper with a summary and conclusions. We also include a short Appendix outlining the calculation of the non-interacting Green's functions in the presence of the on-site core hole potential.

## II.    METHODS

### A.    The Model

Throughout this work, we describe the RIXS process using the Kramers-Heisenberg formalism, where the scattered intensity is directly proportional to the differential cross-section

$$\frac{d^2\sigma}{d\Omega d\omega} \propto \sum_f |\mathcal{F}_{fg}|^2 \delta(E_f - E_g - \omega). \qquad (1)$$

Here, $\mathcal{F}_{fg}$ is the RIXS scattering amplitude

$$\mathcal{F}_{fg} = \sum_{n,i} e^{i\boldsymbol{q}\cdot\boldsymbol{R}_i} \frac{\langle f|D_i^\dagger|n\rangle\langle n|D_i|g\rangle}{E_g - E_n + \omega_{in} + i\Gamma}, \qquad (2)$$

where $E_g$, $E_n$, and $E_f$ are the energies of the initial $|g\rangle$, intermediate $|n\rangle$, and final $|f\rangle$ states of the scattering process, respectively; $\omega_{in}$ and $\boldsymbol{k}_{in}$ ($\omega_{out}$ and $\boldsymbol{k}_{out}$) are the energy and momentum of the incoming (outgoing) X-ray, respectively; $\omega = \omega_{out} - \omega_{in}$ and $\boldsymbol{q} = \boldsymbol{k}_{out} - \boldsymbol{k}_{in}$ are the energy and momentum transferred to the system, respectively; and $D_i$ is a local dipole operator describing the relevant core-valence transition. To be consistent with Ref. 15, we neglect the orbital-dependent factors appearing in the dipole operator and set $D_i = \sum_\sigma d_{i,\sigma}^\dagger p_{i,\sigma}$, where $p_{i,\sigma}$ annihilates a spin $\sigma$ core electron at site $i$ and

$d_{i,\sigma}^\dagger$ creates a spin $\sigma$ valence electron at the same site. We set $\hbar = 1$ throughout this work.

We now consider the case where the incident X-ray locally excites a core electron into an otherwise empty valence band (e.g., a $p \rightarrow d$ transition, although the specific orbitals involved are irrelevant within our level of modeling). To model the $e$-p interactions in the valence band, we use the Holstein Hamiltonian $\mathcal{H} = \mathcal{H}_t + \mathcal{H}_p + \mathcal{H}_{e\text{-}p}$, where

$$\mathcal{H}_t = -t\sum_{\langle ij\rangle}(d_i^\dagger d_j + \text{H.c.}) = \sum_{\boldsymbol{k}}\epsilon_{\boldsymbol{k}}d_{\boldsymbol{k}}^\dagger d_{\boldsymbol{k}}, \qquad (3)$$

describes the hopping of the valence electron with the bare dispersion $\epsilon_{\boldsymbol{k}}$. We focus our discussion on the square lattice with $\epsilon_{\boldsymbol{k}} = -2t\left[\cos(k_x a) + \cos(k_y a)\right]$, where $t$ is the nearest neighbor hopping integral. Generalizations to other band structures and other dimensions are straightforward, and will be mentioned later. As discussed above, $d_i^\dagger$ creates a valence electron on site $i$; we suppress the spin index of the fermion operators from now on as it is irrelevant when at most one valence electron may exist in the system. For this same reason, electron correlations within the valence band are irrelevant and hence ignored.

The second term in the Hamiltonian,

$$\mathcal{H}_p = \omega_0 \sum_i b_i^\dagger b_i, \qquad (4)$$

describes an optical Einstein phonon mode, and the third term,

$$\mathcal{H}_{e\text{-}p} = M \sum_i d_i^\dagger d_i (b_i^\dagger + b_i), \qquad (5)$$

describes the $e$-p interaction between the valence electron and a local phonon mode. Here, $b_i^\dagger$ creates a phonon with energy $\omega_0$ at site $i$, and $M$ is the Holstein $e$-p coupling.

In the intermediate state, the valence electron feels a strong attractive potential from the localized core-hole left behind during the RIXS process. We model this using a local potential

$$\mathcal{H}_{e\text{-}h} = -U_Q \sum_i d_i^\dagger d_i (1 - p_i^\dagger p_i) \qquad (6)$$

as is commonly done in the literature, where $U_Q$ characterizes the strength of the core-hole's potential. (Longer range potentials can easily be treated in a similar manner.)

Our model is identical to the local model used in Ref. 15 for Holstein coupling when the hopping integral $t$ of the $d$-band vanishes (no itinerancy in the intermediate state). We note that because our approach assumes that the core electron is excited into an otherwise empty band, this same electron must ultimately decay and annihilate the core hole. Therefore, we are modeling an indirect RIXS process, similar to Ament *et al.* [15].

Finally, a Holstein coupling of the core-hole to the same phonons, like suggested by in Ref. [17], can be included

in this model with an additional term:

$$\mathcal{H}_{\text{h-p}} = M_h \sum_i (1 - p_i^\dagger p_i)(b_i^\dagger + b_i).$$

We discuss its relevance next.

### B. Coupling the core-hole to the lattice

We begin by examining qualitatively the effects of adding the coupling between the lattice and the core-hole. Let site $i$ be the site where the core hole is created. In the intermediate state, the core hole-phonon Hamiltonian at this particular site becomes: $\mathcal{H}_{\text{p}} + \mathcal{H}_{\text{h-p}} = \omega_0 b_i^\dagger b_i + M_h(b_i^\dagger + b_i) = \omega_0 B_i^\dagger B_i - M_h^2/\omega_0$, where we have introduced the displaced phonon operators $B_i = b_i + M_h/\omega_0$. The e-p coupling at this site now becomes: $\mathcal{H}_{\text{e-p}} = M d_i^\dagger d_i(b_i^\dagger + b_i) = M d_i^\dagger d_i(B_i^\dagger + B_i - 2M_h/\omega_0)$. In other words, using the displaced phonon operators $b_j \to B_j = b_j + \delta_{ij} M_h/\omega_0$, this Hamiltonian maps directly onto a model where the core-hole is not coupled to the lattice, provided that we renormalize the core-hole attractive potential $U_Q \to U_Q^{\text{eff}} = U_Q + 2M M_h/\omega_0$, and shift the overall energy of the intermediate state by $-M_h^2/\omega_0$.

The energy shift in the intermediate state reflects the polaron formation energy associated with the lattice distortion induced by the core-hole. It is an overall constant that shifts the energies $E_n$ of all the intermediary states, so it does not affect the shape of the RIXS spectra.

Much more relevant is the core-hole potential renormalization $U_Q^{\text{eff}} = U_Q + 2M M_h/\omega_0$, which should reduce $U_Q^{\text{eff}}$. We expect a reduction here because $M M_h < 0$ due to the opposite charges of the core hole and the valence electron, which drive opposite distortions of the lattice: if one induces a local expansion, the other induces a local contraction. (In more complex models with multiple bands one can envision other possible scenarios, but for the minimal model studied here, this is the only option). The energy $-2M M_h/\omega_0$ represents an effective on-site repulsion between the core-hole and the valence electron, due to their coupling to the lattice. If the two charges are at different sites, each forms a polaron by creating its optimal local lattice distortion, thus lowering the total energy by $E_{\text{apart}} = -M^2/\omega_0 - M_h^2/\omega_0$ (if we set $t = 0$). In contrast, when the core hole and excited electron occupy the same site, they sabotage each other's distortions and the polaron formation energies are mostly lost. The exciton-polaron energy (again, if $t = 0$) is $E_{\text{onsite}} = -(M + M_h)^2/\omega_0 = E_{\text{apart}} - 2M M_h/\omega_0 \approx 0$ if $M \approx -M_h$, reflecting the fact that a site hosting both the core-hole and the valence electron (i.e. an exciton) is effectively neutral and thus will not distort the lattice much. The valence electron and the core hole occupy atomic orbitals with different wavefunctions so $|M| \neq |M_h|$, although if one envisions the distortion as arising from breathing-mode like displacements of the neighboring O atoms, as is often the case in oxides, then their magnitudes could be rather comparable.

The conclusion is that coupling of the core-hole to the lattice is likely to further undermine the validity of the single-site approximation, because it effectively weakens the core-hole potential from its bare value, thus favoring itinerancy of the valence electron. The more time the valence electron spends at other sites, the more likely it is to create a distortion at those sites and thus leave behind phonons away from the core-hole site. As we show in the following, this leads to a $\boldsymbol{q}$ dependence of the RIXS spectra that is entirely missed by the single-site approximation. This physics becomes more relevant in the limit of stronger e-p and core-hole-phonon couplings.

Determining the absolute importance of the coupling between the core-hole and the lattice requires accurate calculations for $M_h$ and $M$. This task is non-trivial, because the core hole and excited valence electrons will experience different degrees of electronic screening [15]. Nevertheless, this issue should be kept in mind when interpreting RIXS data. This being said, our goal here is to assess the role of electron mobility relative to the purely local model, which neglects this core-hole coupling. We therefore set $M_h = 0$ from here onward. To first order, one can estimate the contribution of the hole-lattice coupling by replacing $U_Q$ with $U_Q^{\text{eff}}$ in the following results. In view of these arguments, we will consider a range of $U_Q$ values in this work that skews *below* those typically found in the literature. For example, $U_Q$ is often taken to be in the range $\approx 4 - 6$ eV [8, 41–43], depending on the elemental edge. Adopting $\Omega \approx 100$ meV as a typical optical phonon energy probed by RIXS, we consider $U_Q/\Omega$ in the range of $10 - 40$ throughout.

A more detailed analysis of both this core hole-lattice coupling and of various other possible generalizations mentioned in the introduction will be presented elsewhere.

### C. MA solution

To apply the MA method, we first recast the scattering amplitude of Eq. (2) in terms of a propagator:

$$F_{fg} = \sum_i e^{i\boldsymbol{q}\cdot\boldsymbol{R}_i} \langle f|p_i^\dagger d_i \mathcal{G}(z) d_i^\dagger p_i|g\rangle, \tag{7}$$

where $\mathcal{G}(z) = [z - \mathcal{H}]^{-1}$ is the resolvent operator, and $z = \omega_{\text{in}} + E_g + i\Gamma$. From now on, we take the initial state $|g\rangle = |0\rangle$ to be the vacuum of excitations (no phonons, no core hole, no valence electron) so that $E_g = 0$.

Following common practice, we assume that the core hole is immobile, so its only role is to provide an on-site attraction $U_Q$ when the valence electron is at the core-hole site $i$: $\mathcal{H}_{\text{e-h}} \to -U_Q d_i^\dagger d_i$. As a result, we need to calculate propagators of the form $\langle f|d_i \mathcal{G}_i(z) d_i^\dagger|0\rangle$, where $|f\rangle$ can have arbitrary numbers of phonons left behind after the core hole decays. From now on we indicate the location of the core hole using the index $i$ of the resolvent $\mathcal{G}_i(z)$, to reflect the attractive potential $-U_Q d_i^\dagger d_i$ included in the Hamiltonian of the valence electron.

Such Green's functions for the valence electron in the presence of this "impurity potential" and of Holstein coupling to the lattice have been calculated with MA for the case $|f\rangle = |0\rangle$ in Refs. [35, 36], where the accuracy of MA was also demonstrated by comparison with state-of-the-art unbiased numerical results. For completeness, we briefly review this solution here and show its generalization for other phonon states $|f\rangle$ compatible with the variational space we use to implement MA. We also review the physical meaning of the approximations made within MA, so that it becomes clear what processes are included and what processes are ignored by it.

We define:

$$G_{ij}^{(i)}(z) = \langle 0|d_i \mathcal{G}_i(z) d_j^\dagger|0\rangle, \qquad (8)$$

where the superscript $(i)$ indicates that the core-hole attraction is at site $i$ (in Refs. [35, 36], the attractive potential is placed at site 0). To calculate this, we employ Dyson's identity:

$$\mathcal{G}_i(z) = \mathcal{G}_{0,i}(z) + \mathcal{G}_i(z)\mathcal{H}_{\text{e-p}}\mathcal{G}_{0,i}(z), \qquad (9)$$

where $\mathcal{G}_{0,i}(z) = [z - (\mathcal{H}_t + \mathcal{H}_{\text{p}} - U d_i^\dagger d_i)]^{-1}$ is the resolvent in the presence of the core-hole potential but in the absence of the $e$-p coupling. This Hamiltonian can be diagonalized straightforwardly so its propagators

$$G_{ij}^{(0,i)}(z) = \langle 0|d_i \mathcal{G}_{0,i}(z) d_j^\dagger|0\rangle$$

are simple to calculate; for completeness, their derivation is provided in the Appendix.

Using Dyson's identity in Eq. (8) we find:

$$G_{ij}^{(i)}(z) = G_{ij}^{(0,i)}(z) + M \sum_l F_1^{(i)}(z,l) G_{lj}^{(0,i)}(z), \quad (10)$$

where for $n \geq 1$ we introduce the generalized propagators:

$$F_n^{(i)}(z,l) = \langle 0|d_i \mathcal{G}_i(z) d_l^\dagger (b_l^\dagger)^n|0\rangle. \qquad (11)$$

Eq. (10) is exact, and reflects the fact that the valence electron can move from site $j$ to any other site $l$ and create a phonon there through the local Holstein $e$-p coupling, hence the appearance of the generalized propagator with $n = 1$. Solving Eq. (10) requires knowledge of all $F_1^{(i)}(z,l)$. We use Dyson's identity to generate equations of motion for these new propagators:

$$F_1^{(i)}(z,l) = \sum_p \langle 0|d_i \mathcal{G}_i(z) \mathcal{H}_{\text{e-p}} d_p^\dagger b_l^\dagger|0\rangle G_{pl}^{(0,i)}(z - \omega_0).$$

Consider $\mathcal{H}_{\text{e-p}} d_p^\dagger b_l^\dagger|0\rangle$. If $p = l$, the $e$-p coupling can either remove the phonon or add a second one. If $p \neq l$, only addition of a second phonon is possible, resulting in propagators with kets of the form $d_p^\dagger b_p^\dagger b_l^\dagger|0\rangle$. Extensive work has shown that at low energies and for Holstein coupling, the latter processes are much less likely than the former [35, 36]: because the coupling is local, it is

energetically favorable for the electron to remain with its phonon cloud once it starts building it rather than abandon it to move elsewhere to form another cloud.

The simplest version of MA, which we implement here, solves the problem within the variational space that only allows single-site phonon configurations like $(b_l^\dagger)^n|0\rangle$. Generalizations to larger variational spaces, where phonons are spread over several sites, are possible and have been implemented for other couplings where they are necessary to obtain accurate results; however, using this specific variational space has been shown to be a very accurate approximation for the Holstein coupling [35, 36]. This approximation reduces the generalized propagators that appear in the equations of motion to only those defined in Eq. (11):

$$F_n^{(i)}(z,l) = M[F_{n+1}^{(i)}(z,l) + n F_{n-1}^{(i)}(z,l)] G_{ll}^{(0,i)}(z - n\omega_0).$$

This recursive equation admits the solution

$$F_n^{(i)}(z,l) = A_n(l-i,z) F_{n-1}^{(i)}(z,l),$$

where the coefficients are the continued fractions

$$A_n(l-i,z) = \frac{n M G_{ll}^{(0,i)}(z - n\omega_0)}{1 - M G_{ll}^{(0,i)}(z - n\omega_0) A_{n+1}(l-i,z)}. \quad (12)$$

They depend only on the distance between the core-hole site $i$ and the cloud site $l$, as expected on physical grounds (mathematically, this is because $G_{ll}^{(0,i)}(z)$ depends on $l-i$, see Appendix A) and are calculated by imposing the physical condition $A_{N_{max}}(l-i,z) = 0$ for a sufficiently large $N_{max}$. (The cutoff $N_{max}$ is found by increasing its value until the continued fractions are converged.) Once these continued fractions are known, we have:

$$F_n^{(i)}(z,l) = \prod_{k=1}^n A_k(l-i,z) G_{il}^{(i)}(z). \qquad (13)$$

Using this in Eq. (10) allows us to bring it into a self-consistent form:

$$G_{ij}^{(i)}(z) = G_{ij}^{(0,i)}(z) + M \sum_l G_{il}^{(i)}(z) A_1(l-i,z) G_{lj}^{(0,i)}(z). \qquad (14)$$

For a finite-size system this equation can be solved as is, but for an infinite system it becomes intractable due to the infinite sum over $l$. Sites $l$ far from the core-hole site $i$ can be efficiently dealt with by noting that $A_1(l-i,z) \to A_1(z)$ when $|l-i| \gg 1$, where

$$A_n(z) = \frac{n M G_{ll}^{(0)}(z - n\omega_0)}{1 - M G_{ll}^{(0)}(z - n\omega_0) A_{n+1}(z)}. \qquad (15)$$

are the corresponding continued fractions in a clean system, with $U_Q = 0$. We define the effective potential

$$v(l-i,z) = M[A_1(l-i,z) - A_1(z)],$$

which goes to zero fast as $l$ moves away from $i$, and rewrite Eq. (14) as

$$G_{ij}^{(i)}(z) = G_{ij}^{(0,i)}(\zeta) + \sum_l G_{il}^{(i)}(z)v(l-i,z)G_{lj}^{(0,i)}(\zeta), \quad (16)$$

with $\zeta = z - MA_1(z)$. A diagrammatic expansion of this effective potential is available in Fig. 2 of Ref. [36], and reveals that it describes the scattering of the electron on the core-hole potential in the presence of the phonons from the polaron cloud.

If $U_Q = 0$, there is no core-hole attraction and this reduces $G_{ij}^{(i)}(z) \rightarrow G_{ij}(z) = G_{ij}^{(0)}(\zeta)$, showing that the free-particle propagator energy is renormalized by the MA self-energy $\Sigma(z) = MA_1(z)$ [19, 20]. This renormalization is responsible for the emergence of the Holstein polaron as the low-energy quasiparticle in the clean system.

If $U_Q \neq 0$, Eq. (16) shows that the full solution has two components. The first is $G_{ij}^{(0,i)}(\zeta)$, which is the bare particle propagator in the presence of the core-hole potential, with the same renormalization of its energy $z \rightarrow \zeta$. It, therefore, can be interpreted as the polaron propagator in the presence of the bare core-hole potential $U_Q$. Interestingly, this is not the full answer. The second part $\sum_l G_{il}^{(i)}(z)v(l-i,z)G_{lj}^{(0,i)}(\zeta)$ shows that the $e$-p coupling also renormalizes this bare core-hole potential by generating an additional potential $v(l-i,z)$. Its dependence on the energy $z$ reflects retardation effects. Unlike the bare core-hole potential, $v(l-i,z)$ is not local although it vanishes fast with increasing $|l-i|$. Previous work [35, 36] shows that we achieve convergence by summing up to second nearest-neighbors (nn) of site $i$, i.e. by imposing a cutoff $p_v = 2$ such that we set $v(l-i,z) \equiv 0$ for $|l-i| > p_v$, in Eq. (16). A more technical discussion of Eq. (16), including a diagrammatic analysis, is provided in Ref. [36]. A more physical discussion is provided below.

If we set $p_v = 2$, Eq. (16) allows us to find $G_{il}^{(i)}(z)$ when $l = i$, $l$ is nn to $i$, and $l$ is a nnn to $i$ from a linear system of three coupled equations. Once these particular $G_{il}^{(i)}(z)$ are known, Eq. (16) produces any other $G_{ij}^{(i)}(z)$ of interest. Other values of the cutoff $p_v$ are handled similarly.

Returning to the RIXS amplitudes of Eq. (7), when $|f\rangle = |0\rangle$ the needed propagator is $G_{ii}^{(i)}(z)$, which is computed as just described. This discussion also shows that the only other final states possible within this variational space are of the form $|f\rangle \propto (b_l^\dagger)^n|0\rangle$, where now $l$ can be *any* site in the system, not just the core-hole site $i$ as assumed in the single-site approximation. This aspect requires us to find the propagators $\langle 0|\frac{b_l^n}{\sqrt{n!}}d_i\mathcal{G}_i(z)d_i^\dagger|0\rangle$,

which are Fourier transforms of the amplitude of probability that after creating a cloud at site $l$, the valence electron returns to site $i$ where it annihilates the core-hole. We remind the reader that we assume that there is a single electron in the valence band in the intermediate state. If the valence band is instead partially filled, then any other electron could decay and fill the core hole, further complicating the analysis.

The new propagators can be linked to the ones already calculated. Let

$$\tilde{F}_n^{(i)}(z,l) = \langle 0|d_i\mathcal{G}_i(z)d_i^\dagger(b_l^\dagger)^n|0\rangle \quad (17)$$

so that

$$\langle 0|\frac{b_l^n}{\sqrt{n!}}d_i\mathcal{G}_i(z)d_i^\dagger|0\rangle = \frac{[\tilde{F}_n^{(i)}(z^*,l)]^*}{\sqrt{n!}}.$$

Within the MA variational space, we find the equation of motion for this propagator to be:

$$\tilde{F}_n^{(i)}(z,l) = M[F_{n+1}^{(i)}(z,l) + nF_{n-1}^{(i)}(z,l)]G_{li}^{(0,i)}(z-n\omega_0).$$

Comparing with the equation of motion for $F_n^{(i)}(z,l)$, we conclude that

$$\tilde{F}_n^{(i)}(z,l) = \frac{G_{li}^{(0,i)}(z-n\omega_0)}{G_{ll}^{(0,i)}(z-n\omega_0)}F_n^{(i)}(z,l)$$

and therefore [see Eq. (13)]:

$$\tilde{F}_n^{(i)}(z,l) = \frac{G_{li}^{(0,i)}(z-n\omega_0)}{G_{ll}^{(0,i)}(z-n\omega_0)}\prod_{k=1}^n A_k(l-i,z)G_{il}^{(i)}(z). \quad (18)$$

All the components in this expression have already been calculated in the process of obtaining the various $G_{il}^{(i)}(z)$.

Within this version of MA, the RIXS cross-section can be expressed as

$$\frac{d^2\sigma}{d\Omega d\omega} \propto \Big|\sum_i e^{i\boldsymbol{q}\cdot\boldsymbol{R}_i}G_{ii}^{(i)}(z)\Big|^2\delta(\omega)$$

$$+ \sum_{n=1}^\infty \frac{1}{n!}\sum_l\Big|\sum_i e^{-i\boldsymbol{q}\cdot\boldsymbol{R}_i}\tilde{F}_n^{(i)}(z^*,l)\Big|^2\delta(\omega-n\omega_0). \quad (19)$$

Using various symmetries such as the fact that $G_{ii}^{(i)}(z)$ is independent of $i$, while $\tilde{F}_n^{(i)}(z^*,l)$ depends only on $i-l$, we can further simplify this to find our final result for the RIXS cross-section:

$$\frac{d^2\sigma}{d\Omega d\omega} \propto \big|G_{ii}^{(i)}(z)\big|^2 N\delta_{\boldsymbol{q},0}\delta(\omega) + \sum_{n=1}^\infty \frac{1}{n!}\Big|\sum_\delta e^{i\boldsymbol{q}\cdot\boldsymbol{R}_\delta}\tilde{F}_n^{(i)}(z^*,i+\delta)\Big|^2\delta(\omega-n\omega_0) \quad (20)$$

where $N \rightarrow \infty$ is the number of sites in the system. In

principle, the sum over $\delta$ in the above expression extends

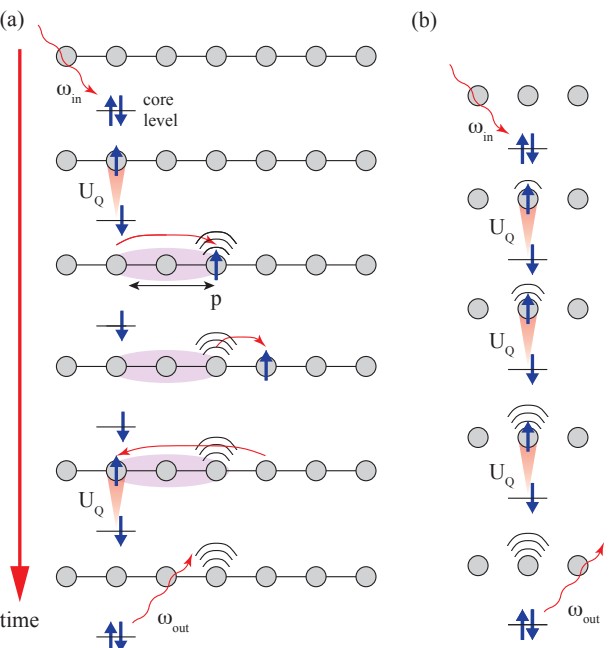

FIG. 1. A schematic representation of the RIXS process as treated in (*a*) the MA theory and (*b*) the Lang-Firsov localized theory.

over all the sites in the system; however, in reality, it converges fast with increasing $|\delta|$ because it is not very probable that phonons will be left behind at sites $i + \delta$ very far away from the core-hole site $i$ after the RIXS process. We define a second cutoff $p$ to truncate this sum, by restricting it to only $|\delta| \leq p$. To simplify the analysis, in the following we set $p = p_v$ and increase its value until convergence is achieved, thus guaranteeing that both cutoffs are sufficiently large. We emphasize that the physical origin of these two cutoffs is very different (as further discussed below) and, therefore, convergence could be achieved for very different values of $p$ and $p_v$ for other models. In those cases, it might be more efficient to converge them separately.

To summarize, the steps in the calculation are (i) choosing the cutoffs for $\delta$ in Eq. (20), for $p = |l - i|$ in Eq. (16), and for $N_{max}$ for the continued fractions of Eqs. (12) and (15). Specifically, we increase these cutoffs until convergence is achieved; (ii) calculating the needed generalized propagators $\tilde{F}_n^{(i)}(z, l)$ of Eq. (18). Specifically, expressions for the bare propagators $G_{li}^{(0,i)}(z - n\omega_0)$ are provided in Appendix, the continued fractions $A_k(l - i, z)$ are obtained from Eq. (12), and the propagators $G_{il}^{(i)}(z)$ are obtained from solving the coupled Eqs. (16); (iii) the expression of Eq. (20) is then evaluated.

The last issue is the choice of $\omega_{\text{in}}$ that enters the $z$ argument when calculating the RIXS intensity. In experiments, this is measured as $\omega_{\text{in}} = \omega_{\text{in}}^{\max} + \Delta$ where $\Delta$ is the detuning from the value $\omega_{\text{in}}^{\max}$ corresponding to the maximum in the x-ray absorption spectroscopy (XAS) spectrum. Within our framework, the XAS intensity can be computed using

$$I_{\text{XAS}} \propto -\frac{1}{\pi} \text{Im} \left[ \sum_n \frac{\langle g | D_i^\dagger | n \rangle \langle n | D_i | g \rangle}{E_g + \omega_{\text{in}} - E_n + i\Gamma} \right]$$
$$= -\frac{1}{\pi} \text{Im} G_{ii}^{(i)}(z). \quad (21)$$

When presenting our results, we always consider the RIXS intensity at resonance, i.e., at an incident photon energy $z = z_{\text{res}}$ chosen such that the zero-phonon (elastic) peak has its maximum amplitude. This energy corresponds to $\omega_{\text{in}} = \omega_{\text{in}}^{\max}$. Later, we will also analyze changes in peak intensities as a function of the detuning $\Delta$ away from this $z_{\text{res}}$ value.

Before moving on, it may be helpful to provide a more physical understanding of the MA approximation to clarify which processes it includes. Consider first when the valence electron is created but without creating a core-hole companion (for instance, in an inverse ARPES experiment). This electron can move anywhere in the system because of the finite hopping $t$, but it propagates as a polaron rather than a bare electron due to the $e$-p coupling. In other words, the electron distorts the lattice in its vicinity and will be temporarily trapped within this local potential. In order to hop, it must first absorb all those phonons, then move to another spot, where it creates another cloud and is temporarily trapped in that vicinity, etc. These processes renormalize the polaron dispersion and are fully included within our approximation – indeed, setting $U_Q = 0$ recovers the Holstein polaron MA spectrum. The approximation in describing this phenomenology is that the phonon cloud's spatial extent is limited to one site. As already mentioned, this is an excellent approximation for the Holstein model and can be relaxed to a more extended cloud for other models, when needed.

Adding the core-hole changes this picture because its attractive potential is likely to keep the polaron closer to the core-hole site. The electron can still travel anywhere with its polaron cloud within our approximation, but the probability of going far from the core-hole site decreases as $U_Q$ increases. However, the presence of the core-hole site has a second effect, which becomes relevant when the polaron cloud is close enough to the core-hole site that this region is sampled by the electron while temporarily trapped within the phonon cloud. In this case, the electron will scatter on the core-hole potential, but this process is affected by the cloud's presence and structure. This renormalization of the effective potential is described by the additional potentials $v(i - l)$ in Eq. (16), when the core-hole is at site $i$ and the polaron cloud is at site $l$. After a time of order $1/\Gamma$ of exploring the lattice in the vicinity of the core-hole site, the electron will recombine with the core-hole, leaving behind its polaron cloud. This cloud can be anywhere in the system but, of course, it is more likely to be near or at the core-hole site.

In contrast to all these processes included within the MA calculation, the one-site approximation restricts the

electron (and therefore its polaron cloud) to only be at the core-hole site, as schematically illustrated in Figure 1.

### D. The Lang-Firsov Localized Limit

If we set $t = 0$, our result simplifies to the well-known one-site formulation, which we will refer to as the Lang-Firsov localized limit. In this case, $G_{il}^{(0,i)}(z) = \delta_{il}/(z+U_Q)$ because the valence electron cannot leave the core-hole site. One consequence of this is that only the $\delta = 0$ term contributes to the $n \geq 1$ peaks of the RIXS cross-section, and its $\boldsymbol{q}$-dependence is lost as a result. The continued fractions become Padé-type expansions in this limit and after some cumbersome work, it can be shown that

$$\mathcal{F}_{fg} \propto \sum_m \frac{B_{n_f m}B_{m n_g}}{z + U_Q - \omega_0(m - \alpha^2)}, \qquad (22)$$

where $n_g$ and $n_f$ are the number of phonons in the initial state $|g\rangle$ and the final state $|f\rangle$, respectively, $\alpha = M/\omega_0$, and

$$B_{mn}(\alpha) = e^{-\alpha^2/2}\sqrt{n!m!}\sum_{l=0}^{n} \frac{(-1)^{m+l}\alpha^{2l+m-n}}{(n-l)!l!(m-n+l)!} \quad (23)$$

are the appropriate Frank-Condon factors for $m \geq n$; for $m < n$ the indices have to be reversed to $B_{nm}(\alpha)$. This is the result predicted by the Ament *et al.* [15] approximation for this model, which we will use for comparison with our MA predictions.

### III. RESULTS AND DISCUSSION

In this section, we present numerical results obtained using the method outlined above and their comparison to the single site solution developed in Ref. [15]. In all cases, we set $\omega_0 = 1$ as the unit of energy. Assuming this is on the order of 100 meV in physical units, we then adopt $t = 5$, $U_Q = 20$–$40$, and $\Gamma = 2$, as representative of the values appearing in the literature [4, 12, 41]. We emphasize that the core hole lifetime broadening $\Gamma$ is responsible for the broadening of the spectral functions along the incident photon energy axis. Separately, we broaden the RIXS spectra as a function of energy loss with a broadening $\eta = 0.05$, so that the $\delta$ functions in Eq. (20) become Lorentzians. This new parameter is mimicking the instrumental broadening related to the resolving power of the monochromator and the detector. These are beyond the theoretical treatment of this work, and so $\eta$ is set to an arbitrary, small number.

In the following we present MA results for different cutoffs $p = 0, 1, 2$. We remind the reader that this cutoff enters the calculation in two different ways: (*i*) it characterizes the range of the renormalized potential, i.e., the range of the sum over $l$ in Eq. (16); and (*ii*) it defines the area where phonons can be left behind after the RIXS

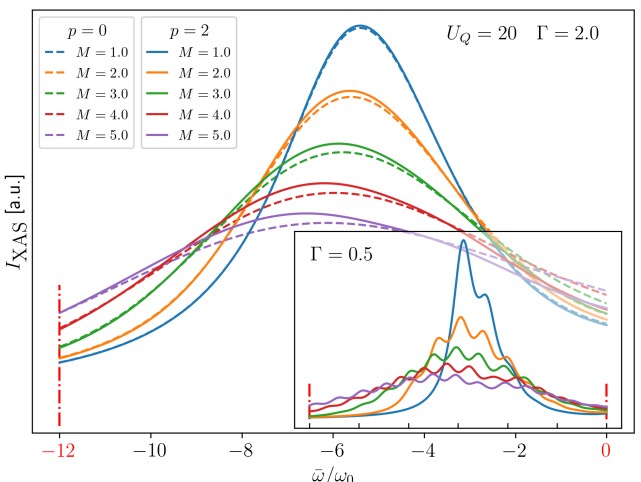

FIG. 2. A comparison of the XAS spectra, as obtained using the MA method for $p = 0$ (dashed lines) and $p = 2$ (solid lines). Results are shown in the main panel for parameters $t = 5, U_Q = 20$, and $\Gamma = 2$. The inset shows results obtained from a $p = 2$ calculation for the same parameter set but with $\Gamma = 0.5$. Note that the x-axis of the inset spans the same range as the main panel.

process, i.e., the sum over $\delta$ in Eq. (20). The latter provides a spatial constraint on the source of the interference effects leading to the $\boldsymbol{q}$ dependence. As mentioned, we could handle the two cutoffs independently but for this model that is not necessary.

### A. Results for the X-ray Absorption Spectra

One important consideration for the theoretical calculations presented herein is the location of the absorption resonance energy at which the RIXS experiment is performed. Generally, the location of the maximum of the XAS spectra, which determines the optimal absorption resonance for the RIXS process, does not coincide with the energy of the non-interacting quasiparticle. For example, the resonance peak is shifted in part by the polaron formation energy (equal to $-M^2/\omega_0$ at $t = 0$), which is already captured in the single-site Lang-Firsov calculation. This shift is also affected by the core hole potential $U_Q$. In experimental practice, this issue is resolved by first performing an XAS experiment to determine the resonance position, and using that value as the input for the RIXS experiment. We perform a similar procedure here, by first finding the XAS maximum from Eq. (21) for any given set of parameters and performing the RIXS calculation at that incidence energy.

Figure 2 presents the XAS spectrum for the parameters $U_Q = 20, \Gamma = 2$ and several intermediate $e$-p coupling values $M$, in a region chosen so as to show where the spectrum undergoes the most dramatic changes. Here, MA results are shown for $p = 0$ and for $p = 2$. They are in good agreement for small couplings $M$, but for larger

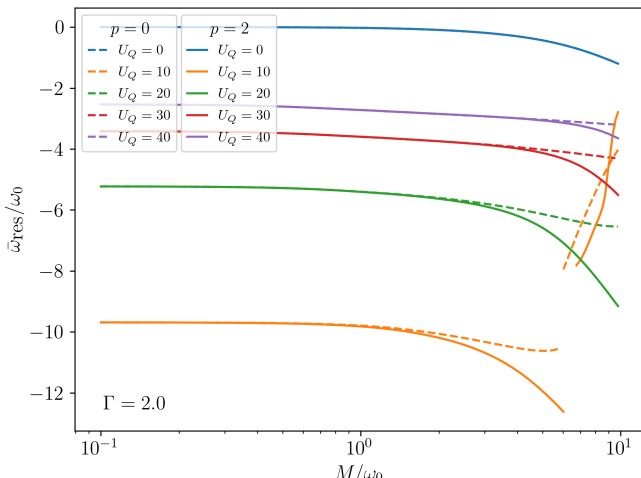

FIG. 3. Resonance position as determined from the maximum of the XAS spectrum, Eq. (21), as a function of $e$-p coupling $M$ and for a wide range of core hole potential values $U_Q$; $t = 5, \Gamma = 2$. The difference between the $p = 0$ (dashed lines) and $p = 2$ (solid lines) solutions is visible at strong coupling $M > 1$. The discontinuity at $U_Q = 10$ is due to bimodality of the XAS spectrum in this regime.

couplings, one can see a difference which indicates that increasing importance of considering final RIXS states with phonons excited away from the core-hole site ($p \neq 0$). The plot is presented as a function of $\bar{\omega} = \Re[z] + U_Q$ to adjust the incidence energy for the core hole potential; thus, $\bar{\omega} = 0$ corresponds to exciting the core electron into the non-interacting quasiparticle state while the other energies correspond to exciting the core electron into a polaronic state where a phonon cloud dresses the carrier. As the coupling constant $M$ increases, the location of the XAS maximum shifts to lower energies, while the spectra broaden and become flatter.

In the inset of Fig. 2 we show the same XAS spectral functions but for a longer core-hole lifetime (smaller broadening $\Gamma = 0.5$) which allows the multi-phonon sidebands, spaced by $\omega_0 = 1$, to be resolved. Their number increases in the strong coupling regime, and turns the XAS absorption into a multi-peak function for which finding the maximum, especially numerically, can become a challenge. This is one reason why we do not consider such small core-hole broadenings in the following.

To further explore the evolution of the XAS spectra, Fig. 3 summarizes the position of its maximum as a function of $M$ and $U_Q$, as determined from the MA calculations $p = \{0, 2\}$. The two MA approximations agree well for small coupling, but they start to deviate from one another for stronger coupling. This is no surprise, as for stronger coupling on average there will be more phonons in the system, and thus higher-order contributions will matter more. What is more surprising is that the weak coupling baseline varies strongly and non-monotonically with $U_Q$: for a wide band system ($U_Q < t$), the resonance is at $\bar{\omega}_{\rm res} \approx 0$. It shifts rapidly to large negative values for

intermediate $U_Q$, but then starts moving back towards zero as $U_Q$ grows. Recall that $\bar{\omega}$ has been corrected for the lowest order shift caused by the core hole potential $U_Q$, so the effect at play here is highly non-linear and resulting from the subtle interplay of parameters in the intermediate regime.

Another interesting feature is that for very strong coupling, $M \gg 1$, $\omega_{\rm res}$ starts to shift more rapidly towards larger negative shifts before turning back towards zero, here exemplified as a discontinuity in the $U_Q = 10$ curve. The physical effects associated with the huge phonon clouds appearing at strong coupling become highly non-linear. This behavior is in contrast to the simple quadratic behavior predicted by the single-site Lang-Firsov treatment. The sudden jump in the $U_Q = 10$ curve is the direct result of the transition to a bimodal spectrum, characteristic of the intermediate core-hole potential $U_Q$, as the intensity of a higher energy sideband gradually overtakes that of the original maximum. Thus, theoretical predictions in this regime should be treated with caution, although the regime deemed physical seems to not be affected by these complications.

We expect that these strong coupling effects observed here are rare and are unlikely to be encountered in most real materials. We will, therefore, focus our attention on studying parameters for which the XAS maintains a single well-defined maximum, and avoid presenting results outside of this regime. In the next section, we first show RIXS results for an incident photon energy set to coincide with this maximum in the XAS spectra. We then explore the dependence of our results on detuning away from this resonance in Sec. III C.

## B. Results for the RIXS Spectra

We begin by highlighting the first of several key, qualitative differences between our results and those of the single site approximation of Ref. [15]. Specifically, we find that the RIXS intensity predicted by the MA approximation with $p \geq 1$ depends on the transferred momentum $\boldsymbol{q}$, even though we are studying (for comparison reasons) a simple model where neither the phonon spectrum nor the electron-phonon coupling have any explicit momentum dependence. This aspect is illustrated in Fig. 4, where we plot the RIXS spectra for MA results corresponding to $p = \{0, 1, 2\}$, for $\boldsymbol{q} = (0, 0), (\pi, 0), (\pi, \pi), (\pi/2, \pi/2)$; all spectra are normalised to the single-phonon peak at $\boldsymbol{q} = (0, 0)$, and curves with different $p$ are shifted vertically to ease comparison.

First, Fig. 4 reveals that the single site ($p = 0$) MA solution has no momentum dependence; however, this is no longer true if $p \geq 1$. The origin of this $\boldsymbol{q}$ dependence stems from the fact that for a finite $p$, there are multiple possible final RIXS configurations, with phonons left behind at different sites within distance $p$ of the core-hole site (these phonons carry the transferred momentum $\boldsymbol{q}$). The total RIXS intensity measures the interference be-

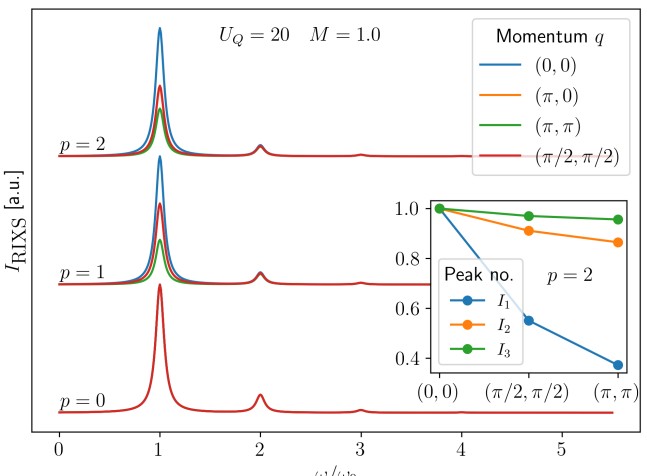

FIG. 4. The RIXS spectrum obtained using the MA method for different $\boldsymbol{q}$ momentum transfers, for increasing cluster sizes $p = \{0, 1, 2\}$. Parameters are $t = 5, U_Q = 20, M = 1, \Gamma = 2, \eta = 0.05$.

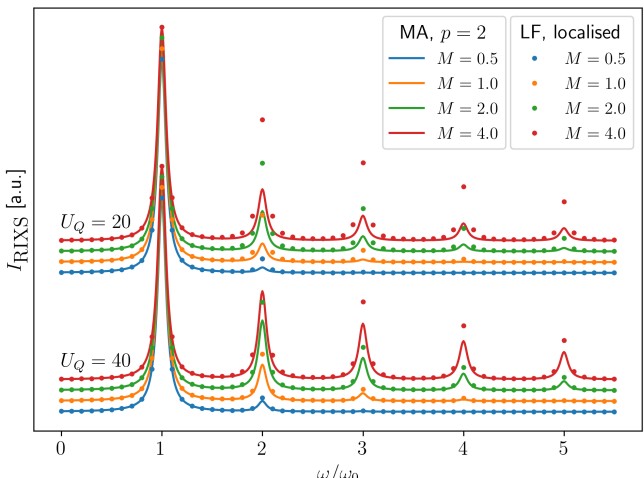

FIG. 5. The RIXS spectrum obtained using the MA method for $p = 2$ (lines, colors for different $e$-p interaction) and the localized Lang-Firsov approximation (points, in corresponding colors). A comparison at $\Gamma = 2$ for $U_Q = 20$ and $U_Q = 40$ is shown; the localized solution effectively corresponds to $U_Q = \infty$. The elastic peak is removed and all data sets are normalized to the one phonon peak for ease of comparison.

tween the amplitudes of probabilities for these various outcomes, see Eq. (20), and therefore depends on $\boldsymbol{q}$. The $p = 0$ MA case, similar to the Ament *et al.* single-site approximation [15], has a single possible final state for a given number of phonons, and thus no interference is possible.

For $\boldsymbol{q} = 0$, the interference is constructive and leads to the highest possible RIXS intensity. The intensity then drops with increasing $\boldsymbol{q}$ – this effect is large and easily visible for the one-phonon peak, but the inset of Fig. 4 shows that it is true for the peaks with more phonons as well. The reason for the suppression of this dependence with increasing $n$ is that the probability for many phonons to be left far from the core-hole site decreases with the number $n$ of phonons, so for larger $n$ the answer is increasingly dominated by the configuration with the phonons at the core-hole site. The same tendency arises (not shown) with increasing $U_Q$, which tends to bind the electron closer to the core-hole site, and thus lower the probability for phonons to be left behind at other sites.

We note that the momentum dependence affects only the intensity of the peaks, but not their locations. This is in agreement with intuition, because the location of the RIXS peaks is controlled by the dispersion of the phonons, and these are dispersionless in the model analyzed here. Our MA approximation is able to treat dispersive phonons, as will be discussed elsewhere, and indeed in that case the peak locations also acquire a $\boldsymbol{q}$-dependence. These two effects provide a nice illustration of how different ingredients of the model may affect different aspects of the RIXS curve, potentially allowing us to identify and quantify them.

Finally, we note that the MA results of Fig. 4 converge fast with increasing $p$, in particular the changes between $p = 1$ and $p = 2$ are very minor. In the following, we will present $p = 2$ results as being essentially converged.

Next, we compare the MA RIXS spectra with those predicted by the single-site approximation. The latter have no $\boldsymbol{q}$ dependence, making a meaningful comparison questionable. We use the $\boldsymbol{q} = 0$, $p = 2$ MA results and scale all spectra to the value of the $n = 1$ peak. The comparison is shown in Figure 5 for two extreme values of the core hole potential $U_Q = 40$ (lower plots) and $U_Q = 20$ (upper plots), for several different $e$-p couplings, marked in different colors. The same spectra, computed using the localized Lang-Firsov approximation of Ref. [15] are also plotted as solid dots for comparison.

As expected, the agreement is better for the larger $U_Q$, which favors the localization of the valence electron closer to the core-hole site. Even there, however, quantitative differences are seen to arise with increasing $e$-p coupling $M$. Specifically, we find that finite $t$ results in a relative decrease of the spectral weight of the multi-phonon excitations, compared to the single-site prediction; however, overall the two methods are in good agreement. In the small $U_Q$ limit, on the other hand, the mobility of the valence electron results in a substantially reduced spectral weight for the multi-phonon peaks, and this is seen even for the weakest $e$-p coupling considered. A way to understand this is that for $t = 0$, the effective $e$-p coupling is essentially infinite, as there is no competing process to the formation of the cloud. A finite $t$, however, brings a second energy scale in the problem: the formation of the phonon cloud (which promotes localization) now competes against the tendency of the valence electron to be in an extended state, promoted by $t$.

Our results show that the effect of the electron mobility on the RIXS spectrum can be substantial. To get a

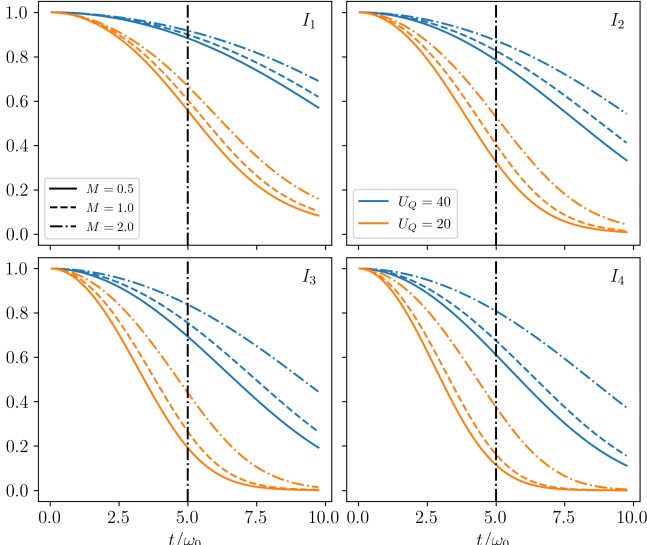

FIG. 6. The amplitude of the first four RIXS peaks at $\boldsymbol{q} = (0,0)$ as a function of the hopping parameter $t$, as calculated using the MA single site method. All curves are normalized to the peak amplitude at $t = 0$. The three sets of parameters under consideration are presented with different colors and the linestyle indicates different $e$-p coupling $M$. The vertical dashed line marks the value of $t$ assumed throughout this work.

better idea of just how important it is, Fig. 6 plots the evolution of the intensity of the first four phonon peaks as a function of $t$, as predicted at $\boldsymbol{q} = 0$ by MA for three different values of $e$-p coupling $M$. All curves are normalized to their corresponding peak intensity at $t = 0$. The value $t = 5$, which is assumed throughout this work, is marked with a dashed line. Fig. 6 clearly shows that the peak intensity decays significantly with increasing $t$ for all relevant $U_Q$ values. Moreover, we can also see that the relative intensity of the higher-order peaks decays faster. This observation is of vital importance for the interpretation of experimental data, as discussed next.

In the seminal work that inspired this research, Ament *et al.* [15] suggested that RIXS might be a particularly useful technique for directly determining the $e$-p coupling strength $M$. This idea is based on a rather involved analysis that consists in plotting, in double-log scale, the ratios of consecutive peaks against the ratio $M/\Gamma$. In the approximation of Ament *et al.* [15], the result is a known function depending *solely on $M/\Gamma$*. This means that if this ratio can be measured experimentally, one can use this known function to infer the corresponding value of $M/\Gamma$. In Fig 7, we plot these corresponding functions for three ratios of consecutive peak intensities, as predicted by Ament *et al.* (curves labeled "LF, localized") [15]. They are seen to be linear over a broad span of $M/\Gamma$ values up to the intermediate values, and saturating towards 1 for larger values. For comparison, we also show our $\boldsymbol{q} = (0,0)$ MA predictions for a mobile valence electron, for $p = 0, 1, 2$ (which again demonstrate that $p = 2$ results

are essentially converged). The results are for the weak core-hole potential $U_Q = 20$, where the effects of the electron itinerancy are more pronounced. Interestingly, while the overall shape of the curves is similar, we see that a finite $t$ produces a significant shift in the curves. This shift might seem like a small change, however, due to the logarithmic scale, it can result in an underestimate of the coupling constant by a factor of 2–3 for these parameters. More importantly, the MA prediction will vary in nontrivial ways as a function of $U_Q$, $t$ and $\boldsymbol{q}$, unlike for the single-site approximation where they only depend on $M/\Gamma$. These observations show that great care has to be taken when analyzing RIXS data using the single site theory, because the effects of electron mobility and the core hole potential can have a significant impact on the final result.

## C. The effects of Electron Mobility on Detuning

Next, we turn our attention to the dependence of the RIXS intensity on the incident photon energy, by allowing it to be detuned away from the XAS resonance, so that $\omega_{\mathrm{in}} = \omega_{\mathrm{in}}^{\max} + \Delta$, where $\Delta < 0$ corresponds to energies below the resonance. Within the single-site model, it has been shown [18] that the detailed shape of these decay curves reveals information about the nature of phononic modes and their coupling strength. To perform a similar analysis, Fig. 8 plots the intensity of the first phonon excitation for several coupling values of $M$ as a function of $\Delta$, again only at $\boldsymbol{q} = (0,0)$. Each curve is normalized to the peak maximum and shifted such that $\bar{\omega} = 0$ to align the curve maxima for comparison. Results are shown for the localized Lang-Firsov theory and for our $p = 0$ and $p = 2$ MA calculation for the weak core-hole potential $U_Q = 20$, where the difference between the theories is more pronounced.

In terms of their relative approximations, the curves obtained using MA in the single-site approximation ($p = 0$) are closest in spirit to those obtained using the single-site Lang-Firsov model. Interestingly, while the trend obtained with both methods looks quite similar overall, the curves decay much faster for the MA treatment of the problem, indicating that the resonance is narrower for a mobile electron. More importantly, we find that the distinction between the different couplings is less prominent within the MA approach, in stark contrast to the Lang-Firsov result, which predicts a strong separation between the curves. At the same time, the difference between the $p = 0$ and $p = 2$ MA calculations is relatively small, which indicates that the extended region for the phonon cloud does not contribute as much as one might expect, given the electron's mobility.

Interestingly, for a stronger core-hole potential ($U_Q = 40$) the picture is slightly different. Although the MA curves still decay faster than those of the single-site theory, and they still differ very little between $p = 0$ and $p = 2$, now they become more clearly separated as a function of $M$, similar to the single-site Lang-Firsov result. The

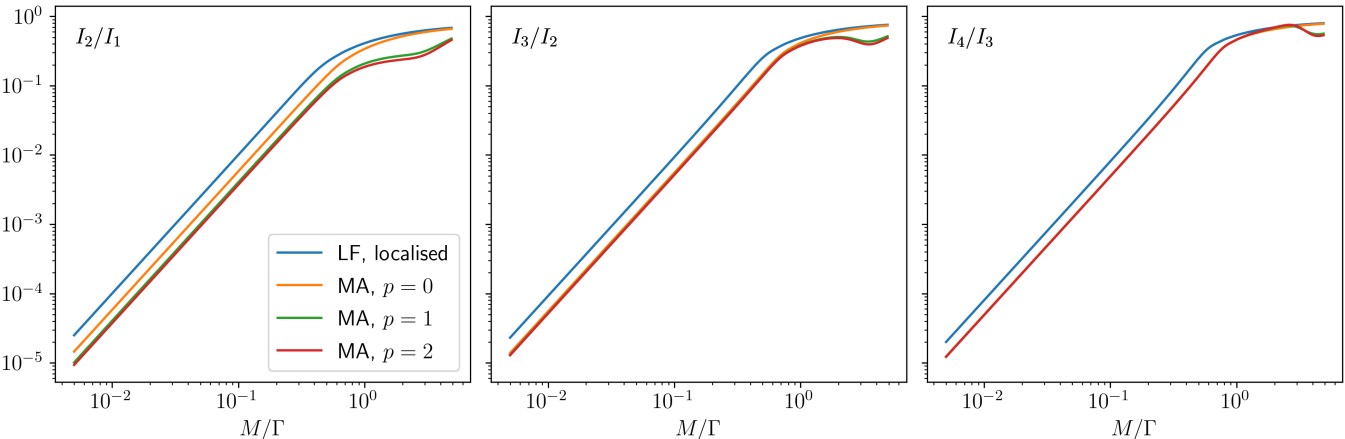

FIG. 7. The first three phonon peak ratios as a function of coupling $M/\Gamma$. The linear dependence implies RIXS might be a good technique for directly determining the $e$-p coupling. Parameters for the O $K$-edge: $t = 5, U_Q = 20, \Gamma = 2$.

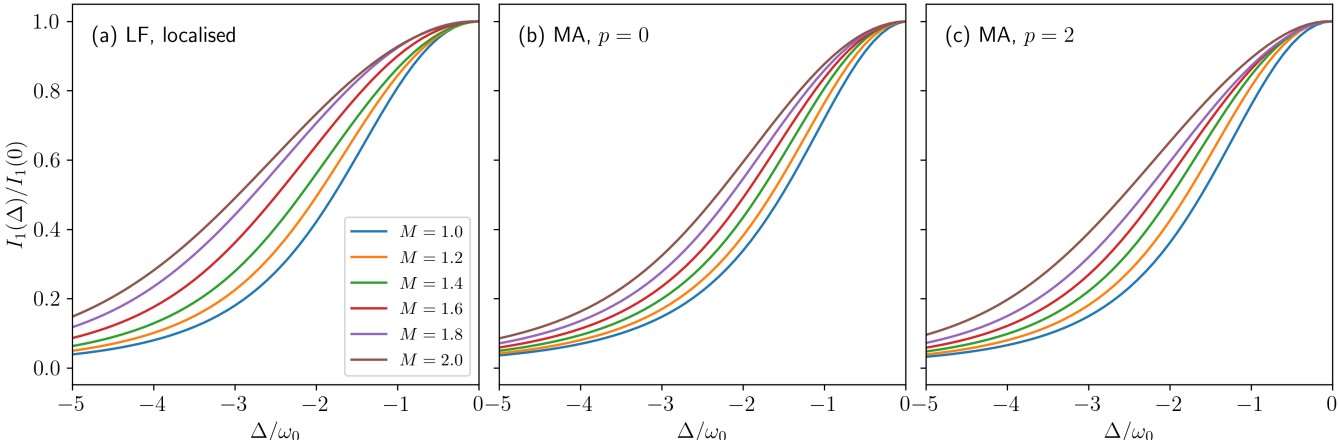

FIG. 8. Single phonon spectral functions $|\mathcal{F}_{1,0}(z)|^2$, corresponding to the amplitude of the one phonon peak, as a function of detuning from resonance; calculated assuming a weak core-hole potential ($t = 5, U_Q = 20, \Gamma = 2$) using (a) the localized Lang-Firsov model, (b) MA $p = 0$, and (c) MA $p = 2$, for a number of different couplings. The curves are shifted and normalised to their maxima to highlight the details of their decay.

reason is that for these parameters the hopping integral $t$ is much smaller compared to the core hole potential, which tends to localize the electron much more than in the previous case.

Thus, we can conclude that while the Lang-Firsov theory could give satisfactory results in specific parameter regimes where the electron mobility can be neglected, it is not sufficient to accurately describe systems where the core hole is long-lived and strongly screened.

### D. The role of dimensionality

Finally, we note that the MA theory presented here differs from the localized Lang-Firsov theory in one more, very fundamental way. The localized theory considers the single core hole site as decoupled from its actual lattice, which makes it effectively a 0D theory. Our MA calcu-

lation is done on an infinite square lattice, and even for $p = 0$ the free-electron propagators retain the information about the 2D band structure of the system. One might, thus, wonder whether the RIXS spectrum is only affected by the bandwidth $W = zt$, or whether the details of the density of states also play a role.

To address this question, Fig. 10 plots for the intensity of the first four phonon excitations as a function of hopping parameter $t$. Here, the intensities are plotted as ratios of values obtained for a 2D square and a 1D lattice. To force both to have the same bandwidth, the 1D hopping parameter is set at twice that of the respective 2D result, $t_{1D} = 2t$. If the results were to depend only on the bandwidth, then the ratios should be equal to one. Indeed, that is the value found for $t = 0$, where both of the models converge on the localized result. In contrast, all the plots display a non-monotonic behavior as a function of $t$, reaching as high as 1.5 before dropping

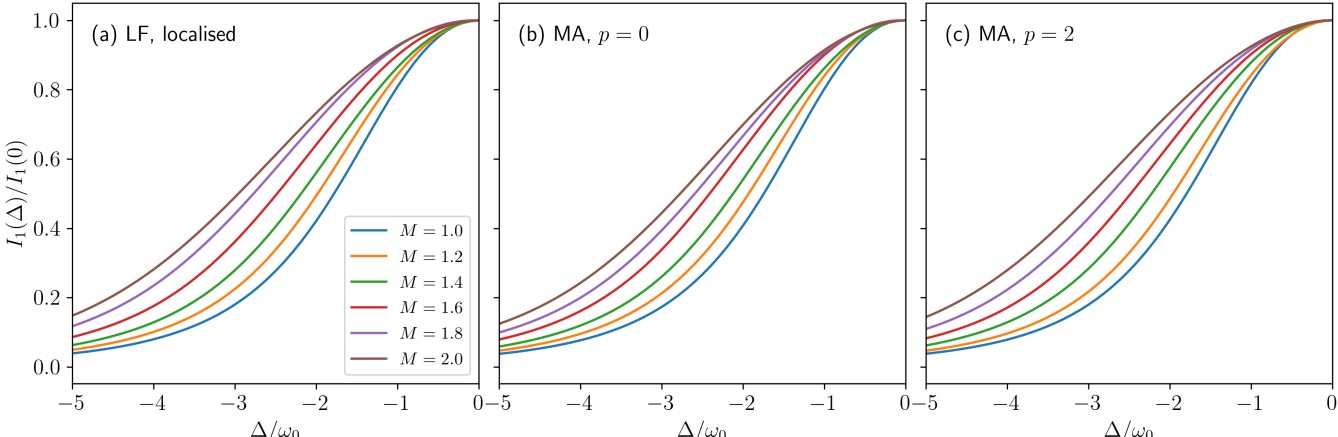

FIG. 9. Single phonon spectral functions $|\mathcal{F}_{1,0}(z)|^2$, corresponding to the amplitude of the one phonon peak, as a function of detuning from resonance; calculated assuming a strong core-hole potential ($t = 5, U_Q = 40, \Gamma = 2$) using (a) the localized Lang-Firsov model, (b) MA $p = 0$, and (c) MA $p = 1$, for a number of different couplings. The curves are shifted and normalised to their maxima to highlight the details of their decay.

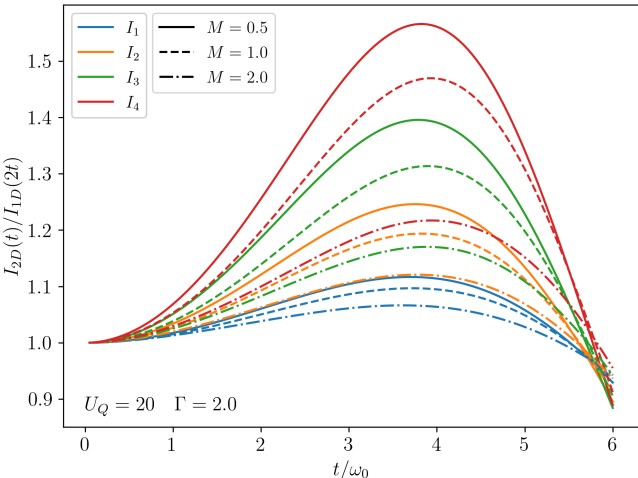

FIG. 10. Ratio of the first four RIXS peaks (denoted with different colors) between the 2D and the 1D model with twice the bandwidth ($t_{1D} = 2t$) and for three different couplings $M$ (indicated with different line types). Calculated with MA $p = 0$ for parameters $U_Q = 20, \Gamma = 2$.

off below one at large $t$. This result demonstrates that intensities of the phonon excitations for the 2D case at first diminish slower than for 1D, but eventually drop below the 1D signal. More importantly, the fact that we observe a strong $t$ dependence indicates that the peak intensities are also sensitive to the dimension of the system and its lattice geometry. These aspects, then, should be added to the long list of other factors (besides $M/\Gamma$) that control the location of the curves like those shown in Fig. 7, and which will therefore influence the $M/\Gamma$ value that one might obtain based on them.

## IV. SUMMARY AND CONCLUSIONS

We have extended the Momentum Average (MA) approximation—a well established, variational, semi-analytical method for computing Green's functions—to model the effect of $e$-p coupling on RIXS spectra. Our approach can be used to treat the case where the core electron is excited into an empty band where it interacts with the lattice, and it improves on the widely used single-site model based on a localized Lang-Firsov formalism in that it allows us to consider the role of electron mobility $t$ and its interplay with the core-hole potential $U_Q$. This is the aspect that we focused on here, even though MA can be generalized to study a much broader class of models.

Using MA, we have demonstrated that the localized model is insufficient for analyzing RIXS data when the valence electron is expected to be more delocalized. Moreover, we showed that the electron's mobility is expected to be particularly crucial at edges with shallow, long-lived core hole states. This result has important implications for future RIXS experiments attempting to extract quantitative estimates for the $e$-p coupling constant from O $K$-edge measurements. For example, our work suggests that the improper use of a fully localized theory severely *underestimates* the electron-phonon coupling obtained from the relative multi-phonon peak analysis.

We also observed several other interesting effects that were not considered before and could not be derived from the localized model. One such effect was the observation that electron mobility induces a dependence on the momentum transfer $q$, even though the $e$-p coupling is completely momentum independent. Another critical point is the strong suppression of the RIXS signal with electron delocalization, due to a competition between mobility and electron-phonon coupling, a fact also with potential experimental consequences. We also show that our results

are sensitive not only to the bandwidth but also to the details of the density of states. Finally, we analyzed the dependence of the resonance position (XAS maximum) on the model parameters, as well as the behavior of the spectral function away from resonance. All of these facts are of empirical importance and offer highly non-trivial opportunities to verify our predictions experimentally.

Finally, we stress that our results are obtained in the limit of a single carrier in the intermediate state. They are, therefore, most relevant to band insulators. At this time it is unclear how much of this will carry over to the many-particle case. It is possible that strong correlations in cuprates, for example, could reduce the importance of itinerancy in the intermediate state. Nevertheless, given the significant *qualitative* changes in the RIXS spectra we have observed here, a theory must be developed for the many-particle case.

## ACKNOWLEDGMENTS

K. B. and M. B. are supported by the UBC Stewart Blusson Quantum Matter Institute (SBQMI) and by the Natural Sciences and Engineering Research Council of Canada (NSERC). S. J. is supported by the National Science Foundation under Grant No. DMR-1842056.

## Appendix A: Free propagators

### 1.  In the clean system

The 2D real space Green's function for the clean lattice (no core-hole potential) is defined as

$$G_{ij}^{(0)}(z) = \langle 0|d_i \mathcal{G}_0(z) d_j^\dagger|0\rangle, \tag{A1}$$

where $\mathcal{G}_0(z) = [z - \mathcal{H}_t]^{-1}$ is the resolvent operator for a free electron. $\mathcal{H}_t$ is trivial to diagonalize and thus the free propagators can be expressed as a Fourier transform of the momentum space Green's function

$$G_{ij}^{(0)}(z) = \frac{1}{(2\pi)^2} \int d^2k \frac{e^{i\boldsymbol{k}\cdot(\boldsymbol{R}_i - \boldsymbol{R}_j)}}{z - \epsilon_{\boldsymbol{k}}}, \tag{A2}$$

where $\epsilon_{\boldsymbol{k}}$ is the 2D electron dispersion. $G_{ij}^{(0)}(z)$ depends only on the relative distance $|\boldsymbol{R}_i - \boldsymbol{R}_j|$, as expected because of invariance to lattice translations.

The above integral can be expressed analytically in terms of elliptic integrals, using a set of recurrence relations [44]. There also exist efficient numerical procedures analogous to the continued fraction technique [45, 46], which allow for accurate calculations of these Green's functions.

### 2.  With the core-hole potential

Consistent with the main text, we define the inhomogeneous Green's function as

$$G_{jl}^{(0,i)}(z) = \langle 0|d_j \mathcal{G}_{0,i}(z) d_l^\dagger|0\rangle, \tag{A3}$$

where $\mathcal{G}_{0,i}(z) = [z - (\mathcal{H}_t + \mathcal{V}_i)]^{-1}$ is the resolvent operator and $\mathcal{V}_i = -U d_i^\dagger d_i$ is the attraction from the core-hole located at site $i$.

Using Dyson's identity, it is straightforward to show that

$$G_{jl}^{(0,i)}(z) = G_{jl}^{(0)}(z) - U G_{ji}^{(0,i)}(z) G_{il}^{(0)}(z),$$

from which we find the propagator to/from the core hole site

$$G_{ji}^{(0,i)}(z) = \frac{G_{ji}^{(0)}(z)}{1 + U G_{ii}^{(0)}(z)},$$

which we can now use to find the general propagator for any pair of sites

$$G_{jl}^{(0,i)}(z) = G_{jl}^{(0)}(z) - U \frac{G_{ji}^{(0)}(z) G_{il}^{(0)}(z)}{1 + U G_{ii}^{(0)}(z)}.$$

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
