# Peer review of "Beyond the single-site approximation modeling of electron-phonon coupling effects on resonant inelastic X-ray scattering spectra"

_SciPost Physics_

## Round 2 · Referee Report · Anonymous (Referee 1) · 2021-7-1

Strengths

  1. Addresses a problem that is key to accurate (especially quantitative) interpretation of RIXS experiments on condensed matter.
  2. Extremely well written, clear and methodical.
  3. Calculations presented are thorough and complete.
  4. Presents a good first step to understanding the importance of itinerancy in RIXS spectra for phonons, allowing for future improvements incorporating more complex effects (phonons with dispersion etc).
  5. Useful example calculations are presented that make clear the differences between this method and existing treatments.

Weaknesses

  1. I think the calculation discussed in section II C. would be clearer if the diagrammatic language version was also presented.
  2. It isn't clear to me what is involved in the numerical work for evaluation of Eq 19 - maybe this is just a case of using numerical methods to evaluate $G$ and the double sum, including the continued fractions? For the latter how is the $N$ for the 'physical condition' that $A_N =0$, for sufficiently large $N$, chosen?
  3. No distinction is made between indirect or direct RIXS. Maybe this isn't needed, though if so it would be good to explain why.
  4. (Minor) The method is described as variational. Typically in a variational method one has a set of parameters that are varied in some way to optimise a solution. What is being 'optimised' in MA?
  5. (Minor) The work considers only dispersionless phonons and focuses on the relative intensities of phonon peaks, as this is what is most usefully compared to previous work based on the single-site (Lang-Firsov) limit. This is important, but I'd have liked some discussion of how the qualitative findings (e.g. the introduction of a q dependence) might in future work influence the (to me) more interesting case of phonons with dispersion.

Report

RIXS has become an increasingly important technique for analysing collective excitations in condensed matter, complementing neutron scattering.
However relative to neutron scattering, RIXS can work with smaller samples and in applied fields; be used for time resolved pump-probe experiments; and study electronic excitations.
The disadvantage relative to neutron scattering is that the physics of the interaction between the necessary core hole excitation and the other excitations in the system makes interpretation of RIXS data more difficult.

For the study of phonon spectra, and electron-phonon interactions, this paper makes an important step by going beyond the current single site approximate treatment to include the effect of mobile electrons in the valence band, and shows that there can be qualitative and quantitative consequences for reasonable parameter regimes. As such I think the paper arguably meets the acceptance criteria expectation 1 (it breaks new ground by showing the importance of mobile valence electrons for RIXS) and also expectation 3: the method presented constitutes a new pathway that will allow for incorporation of further more complex effects, such as phonons with dispersion, etc.

I found the paper very well written, pedagogical and properly self-contained, and it meets all the general acceptance criteria.

There are a few of points below that I'd like the authors to consider.

I thought the treatment in Section II C would have been easier to follow if a couple of figures showing the diagrammatic language were also presented. I understand that Ref 36 might cover this, but having to refer to that makes the manuscript feel less complete.
Equation 8 defines the propagator as $G_{ij}^{(i)}$ where the core hole sits at the same site as the valence annihilation operator $d_i$. However more general propagators (for example $G_{l j}^{(0,i)}$) appear on the RHS of Equations 10 and 16 (and in some unnumbered equations). Maybe it would be better to originally define this more general expression in Equation 8? Instead the more general definition appears in Appendix A 2.

The actual steps involved in the numerical treatment are somewhat opaque (though I expect they would become clearer once one started trying to evaluate Eq 16) so it would be good to have a few sentences explaining the steps at the beginning of the results section.

The model studied is effectively spinless, which makes sense for the aim of this particular work, but can the authors comment on whether an analogous treatment to MA is possible for coupling to magnons instead of phonons?

Requested changes

  1. Consider including diagrammatic accompaniment to the calculation in Sec II C.
  2. Briefly explain the necessary numerical steps at the beginning of the results section.
  3. Explain the (ir)relevance to indirect or direct RIXS.
  4. Make it explicit what is variational in the MA method, and what quantity is optimised.
  5. Consider using the more general propagator definition based on the form in Equation (A3) in Section II C.
  6. Define XAS on page 6 when it first appears (instead it is defined on page 8).
  7. There are. few places (page 7 and 9) where the 'Ament et al.' approximation is referred to, but without explicitly giving the ref [15].
  8. Possibly a typo - at the beginning of II B "the core hole-phonon Hamiltonian at this particular site becomes $\mathcal{H}_{h-p}=\cdots$" But to me this looks like it should be "$\mathcal{H}p+\mathcal{H}=\cdots$"

  • validity: top
  • significance: high
  • originality: good
  • clarity: top
  • formatting: perfect
  • grammar: perfect

Author:  Krzysztof Bieniasz  on 2021-08-27  [id 1714]

(in reply to Report 1 on 2021-07-01)

RESPONSE TO REFEREE 1

The Referee requested: 1. Consider including diagrammatic accompaniment to the calculation in Sec II C.

Our Response: We agree that having a diagrammatic representation of the processes considered in our variational space could be of use to the reader. However, we note that the specific diagrammatic expansion associated with our level of MA modeling, capturing both polaron propagation and impurity scattering) has been shown in a previous work by one of the authors [see Figs. 1 and 2 of H. Ebrahimnejad and M. Berciu, Phys. Rev. B 86, 205109 (2012).] Providing a similar discussion and figure here might further clarify the nature of the $v(l-i, w)$ potential; however, since this has already been published elsewhere, we feel that repeating this discussion would be inappropriate. Instead, we have added a statement referring the reader to the availability of the diagrammatic analysis in this earlier work.

The Referee requested: 2. Briefly explain the necessary numerical steps at the beginning of the results section.

Our Response: We have added a paragraph below Eq. (20) that summarizes the main steps in calculating the RIXS cross-section.

The Referee requested: 3. Explain the (ir)relevance to indirect or direct RIXS.

Our Response: For direct RIXS, the incoming photon excites the core-electron to an empty valence band state, while a different electron decays and annihilates the core-hole. For indirect RIXS, the same core electron that was excited by the incoming phonon is the one that decays. For technical reasons outlined in our paper, our MA approach applies to the case where the core electron is excited into an otherwise empty valence band. Therefore, only indirect RIXS processes are allowed, since there are no additional electrons available to annihilate the core hole. We agree that this aspect should be clearly stated in the manuscript. In the revised text, we now state explicitly that our approach is currently modeling an indirect RIXS process.

The Referee requested: 4. Make it explicit what is variational in the MA method, and what quantity is optimised.

Our Response: We added a couple of sentences to the introduction to explain that the variational nature of MA is due to the constraints on the allowed carrier and phonon configurations. In other words, instead of solving the problem in the full Hilbert space that contains all possible carrier and phonon configurations, we select a class of configurations that are the most relevant (i.e., have the highest weight in the relevant wavefunctions). This choice is then verified by increasing the variational space until a desired accuracy is achieved.

Previously, we have shown that very good accuracy is achieved for the Holstein model by including only 'one-site cloud' configurations, where the phonons are constrained to all be on the same site, see Refs. [19-21] of the main paper. (There is no constraint in the number of phonons, or in how far this site is located from the carrier.) We consider the same variational space in this work. A systematic way to increase it is to also allow configurations where phonons are spread over several sites (i.e., the 'cloud' is more extended, like in Refs. [22-24]). Again, for the Holstein model, with its purely local el-ph coupling, it turns out such extended cloud configurations are not relevant except in the extremely adiabatic limit. For other models, however, more extended clouds are relevant even in the anti-adiabatic limit, so care needs to be taken to implement a correct MA variational space, consistent with the problem studied.

The Referee requested: 5. Consider using the more general propagator definition based on the form in Equation (A3) in Section II C.

Our Response: The more general propagators $G^{(i)}_{lj} (z)$ can indeed be calculated; however, they are not needed for finding the RIXS cross-section. Only the $G^{(i)}_{ij}(z)$ propagators appear there, as we have explained in the newly added summary (see our answer to Q2 above). Physically, only $G^{(i)}_{ij}(z)$ enters because the RIXS processes always begin with the valence electron at the core-hole site $i$. By contrast, the more general bare propagators $G^{(0,i)}_{lj}(z)$ are needed (and can be calculated from the expressions given in the Appendix) because the carrier can move between different sites in the system, during any RIXS process. Given this, we believe that it is better to restrict the discussion to only how to calculate the relevant propagators $G^{(i)}_{ij}(z)$ and decided to not generalize this discussion.

The Referee requested: 6. Define XAS on page 6 when it first appears (instead it is defined on page 8).

Our Response: We thank the referee for pointing this out. We have corrected this in the revised text.

The Referee requested: 7. There are few places (page 7 and 9) where the 'Ament et al.' approximation is referred to, but without explicitly giving the ref [15].

Our Response: We have added explicit references to Ament et al.'s paper where they were missing.

The Referee requested: 8. Possibly a typo - at the beginning of II B ''the core hole-phonon Hamiltonian at this particular site becomes $\mathcal{H}_{h-p} = \dots$.'' But to me this looks like it should be "$\mathcal{H}_{p}+\mathcal{H}_{h-p} = \dots$.''

Our Response: The referee is correct, and we have corrected this typo. Thank you for bringing this to our attention.

In addition, the Referee mentioned in the "Weaknesses" section two issues that were not repeated in "Requested changes". We address them here:

The Referee requested: 2. It isn't clear to me what is involved in the numerical work for evaluation of Eq 19 - maybe this is just a case of using numerical methods to evaluate G and the double sum, including the continued fractions? For the latter how is the N for the 'physical condition' that $A_N=0$, for sufficiently large $N$, chosen?

Our Response: The first part was addressed in Q2 above. Regarding the cutoff $N$ (which we renamed $N_{max}$ to avoid confusion with the number of sites $N$), this is increased until the values of the various continuous fractions become converged. Physically, it has to be large enough so that processes involving $N_{max}$ or more phonons are extremely unlikely. We have added some text to clarify this issue.

The Referee requested: 5. (Minor) The work considers only dispersionless phonons and focuses on the relative intensities of phonon peaks, as this is what is most usefully compared to previous work based on the single-site (Lang-Firsov) limit. This is important, but I'd have liked some discussion of how the qualitative findings (e.g. the introduction of a q dependence) might in future work influence the (to me) more interesting case of phonons with dispersion.

Our Response: We are currently finalizing the writing of a manuscript addressing RIXS spectra for dispersive phonons (with Holstein coupling). While full details will be revealed there, one obvious place where the phonon dispersion makes a difference is in the location and width of $n$-phonon peaks appearing in the RIXS spectra, which are no longer Lorentzians located at $n\omega_0$ like for Einstein phonons. For $n=1$, the peak is located at $\omega_q$ (i.e., it tracks the phonon dispersion) and is still a Lorentzian whose width is controlled by the "instrumental broadening". For $n\ge 2$, instead of Lorentzians we find a continuum spanning the convolution of $n$ phonon dispersions corresponding to a total momentum $q$.

Finally, in their Report, the Referee made an excellent suggestion that was not repeated in the "Requested changes", namely:

The Referee said: The model studied is effectively spinless, which makes sense for the aim of this particular work, but can the authors comment on whether an analogous treatment to MA is possible for coupling to magnons instead of phonons?

Our Response: We thank the Referee for bringing this up. Indeed, this approach can be generalized to study magnons instead of phonons, given that we have already demonstrated the success of the variational method in describing spin-polarons in both FM and AFM backgrounds. We added a short paragraph in the Introduction to mention this possible generalization.

---

## Round 2 · Referee Report · Anonymous (Referee 2) · 2021-8-18

Strengths

  1. Clear and well written.

  2. Points out limitations in previous work (Ref. 15).

  3. Text is pedagogical.

Weaknesses

None that are serious. I have requested clarification on a number of points below.

Report

In this manuscript, “Beyond the single-site approximation modeling of electron-phonon coupling effects on resonant inelastic X-ray scattering spectra”, the authors consider the problem of how the electron-phonon coupling manifests itself in a resonant inelastic x-ray (RIXS) measurement. More specifically, the authors want to understand the nature of the single-site approximation employed in the pioneering work of Ament et al (Ref. 15) on how the RIXS response can be used to measure the strength of the electron-phonon coupling.

The single site approximation assumes that the phonons that are created in the RIXS process are localized spatially and only found on the site where the core hole is created. In contrast, the authors use an approach known as the Momentum Average (MA) approach that permits phonons to be created at a distance from the core hole site.

The MA approach itself involves an approximation. It is a variational approach that restricts the Hilbert space of the problem to states that involve some number of phonons of a single type, but states where phonons of different types, i.e., that are found on multiple sites, are not included in the variational space.

Using the MA approach, the authors argue that the single site approximation misconstrues the following features of the RIXS response for a material with electron-phonon coupling:

i) The single site approximation leads to a “severe” underestimate of the electron-phonon (e-p) coupling obtained from a multi-phonon peak analysis.

ii) Although the e-p coupling is momentum independent, the itineracy of the valence electron leads to a q-dependent RIXS signal.

iii) When the core hole is relatively long lived, the neglect of itineracy effects is inappropriate.

iv) The RIXS response is sensitive to details of the band structure of the valence electron including its bandwidth and density of states.

The paper is very well written and the results that it arrives at are provide important clarifications on the limitations of the single-site approximation. I thus recommend acceptance.

I do have the following minor comments that the authors might want to consider.

Requested changes

1. What is the effect of electron-electron interactions? Can the authors even give a preview (if only at the Hartree-Fock level) in the conclusions?

2. As an example of a material where the analysis in the manuscript might be applicable, the authors cite work on nSrIrO3/mSrTiO3 heterostructures with nearly empty valence bands. Does this material have the Einstein phononic modes this work suppose? If not, this should be made clear to the reader.

Are there other materials to which the analysis in this paper might be applicable?

3. The authors point out many papers where the MA approximation has been proven to be good. But it would be helpful for the reader to know if this is so for the entire parameter range of the Holstein model. Or are there limitations?

4. What is a reasonable value for the parameter U_Q in a real material such as nSrIrO3/mSrTiO3 heterostructures? Stating this in the text would be a useful guide for the reader in interpreting the presented numerical data.

  • validity: high
  • significance: high
  • originality: high
  • clarity: top
  • formatting: perfect
  • grammar: perfect

Author:  Krzysztof Bieniasz  on 2021-08-27  [id 1715]

(in reply to Report 2 on 2021-08-18)

RESPONSE TO REFEREE 2

The Referee Wrote: In this manuscript, ``Beyond the single-site approximation modeling of electron-phonon coupling effects on resonant inelastic X-ray scattering spectra", the authors consider the problem of how the electron-phonon coupling manifests itself in a resonant inelastic x-ray (RIXS) measurement. More specifically, the authors want to understand the nature of the single-site approximation employed in the pioneering work of Ament et al (Ref. 15) on how the RIXS response can be used to measure the strength of the electron-phonon coupling.

The single site approximation assumes that the phonons that are created in the RIXS process are localized spatially and only found on the site where the core hole is created. In contrast, the authors use an approach known as the Momentum Average (MA) approach that permits phonons to be created at a distance from the core hole site.

The MA approach itself involves an approximation. It is a variational approach that restricts the Hilbert space of the problem to states that involve some number of phonons of a single type, but states where phonons of different types, i.e., that are found on multiple sites, are not included in the variational space.

Using the MA approach, the authors argue that the single site approximation misconstrues the following features of the RIXS response for a material with electron-phonon coupling: i) The single site approximation leads to a ``severe" underestimate of the electron-phonon (e-p) coupling obtained from a multi-phonon peak analysis. ii) Although the e-p coupling is momentum independent, the itineracy of the valence electron leads to a q-dependent RIXS signal. iii) When the core hole is relatively long lived, the neglect of itineracy effects is inappropriate. iv) The RIXS response is sensitive to details of the band structure of the valence electron including its bandwidth and density of states.

The paper is very well written and the results that it arrives at are provide important clarifications on the limitations of the single-site approximation. I thus recommend acceptance.

Our Response: We thank the reviewer for their time, their interest in our work, their positive recommendation, and their valuable comments and suggestions.

The Referee requested: 1. What is the effect of electron-electron interactions? Can the authors even give a preview (if only at the Hartree-Fock level) in the conclusions?

Our Response: At the moment, we do not know how to do a controlled approximation to even begin to answer this question within the MA formalism. The reason is as follows. First, there is the well-known challenge of dealing accurately with (bare) electron-electron ($e$-$e$) interactions. In principle, it might seem that if these are not too strong, one could use Hartree-Fock (as suggested by the Referee) and replace the current free electron propagator with the corresponding HF electron propagator. This can certainly be done, but we believe that its accuracy would be rather dubious. The reason is that the bare $e$-$e$ interactions are supplemented by phonon-mediated effective $e$-$e$ interactions, which need not be weak (if the $e$-ph coupling is not small) and moreover contain retardation effects, etc. In diagrammatic terms, the scheme described above would be equivalent to assuming that bare $e$-$e$ interaction lines do not cross with phonon-mediated $e$-$e$ interaction lines. This might be a reasonably accurate approximation if the time/energy scales for these interactions were very different, but it's quite unlikely to be generically valid.

The problem is that at the moment we do not have a next "better step'' to compare against, to gauge if/when this approximation is reasonable. Absent the "better step'', it is impossible to judge if the results are likely to describe real phenomenology, or are just consequences of a poor approximation. For this reason, we prefer to postpone dealing with the finite carrier concentration problem until we can treat this case in a more controlled fashion, like we do for the single carrier limit in this work. This topic is being actively studied by our groups using more advanced numerical methods.

The Referee requested: 2. As an example of a material where the analysis in the manuscript might be applicable, the authors cite work on $n$SrIrO${}_3$/$m$SrTiO${}_3$ heterostructures with nearly empty valence bands. Does this material have the Einstein phononic modes this work suppose? If not, this should be made clear to the reader. Are there other materials to which the analysis in this paper might be applicable?

Our Response: Indeed, the relevant phonon modes in the $n$SrIrO${}_3$/$m$SrTiO${}_3$ heterostructures are the LO4 optical oxygen modes with energies centered around $\Omega \approx 100$ meV and with dispersion relations varying by 10--20 meV. These phonon branches are often well approximated by Einstein phonons. We have inserted a comment about this in the main text. Regarding other materials, we are not aware of other RIXS studies on band insulators. But as the field of phonon RIXS matures, we believe that such studies will be conducted.

The Referee requested: 3. The authors point out many papers where the MA approximation has been proven to be good. But it would be helpful for the reader to know if this is so for the entire parameter range of the Holstein model. Or are there limitations?

Our Response: Thank you for raising this important issue. Previous work from some of us has shown that MA becomes increasingly less quantitatively accurate in the extreme adiabatic limit (as can be expected, given that there one would expect the appearance of more extended clouds, which are not included within the variational states considered here). However, it was very recently shown that the MA approach can be generalized to arbitrary accuracy by using essentially brute-force numerics to significantly increase the variational space (and computational times, of course). The work presented in newly added Ref. [38] directly mirrors the MA approach, meaning that it can be straightforwardly generalized to calculating the RIXS-relevant propagators as well. We have added a paragraph in the main text to alert the readers to this very recent development, and its possible use for further improving the accuracy of phonon-related RIXS predictions.

The Referee requested: 4. What is a reasonable value for the parameter $U_Q$ in a real material such as $n$SrIrO${}_3$/$m$SrTiO${}_3$ heterostructures? Stating this in the text would be a useful guide for the reader in interpreting the presented numerical data.

Our Response: The answer to this question depends on the elemental edge of the RIXS experiment. The $n$SrIrO${}_3$/$m$SrTiO${}_3$ experiments we're citing were conducted at the oxygen $K$-edge, where the core-hole potential is typically taken to be $U_Q = 4 - 6$ eV. Assuming that $\Omega = 0.1$ eV, this would translate into $U_Q = 40 - 60$ in our units where $\Omega$ is the unit of energy. However, we stress that our analysis in Sec. IIB also showed that the ``effective'' core hole potential is reduced by any additional core hole-lattice coupling in the system.

Determining the strength of this additional coupling requires more detailed calculations beyond the scope of this work. Nevertheless, these effects should reduce the value of $U_Q$ in the system and this is reason why we also considered a range of $U_Q \approx 10$--40 in our work.

In response to the Reviewer's comment, we have inserted some additional explanation along these lines when we introduce the relevant parameter ranges.

---

## Editorial Decision

resubmitted